# Major urinary protein (*Mup*) gene family deletion drives sex-specific alterations in the house-mouse gut microbiota

Madalena V. F. Real,[1] Melanie S. Colvin,[2] Michael J. Sheehan,[2] Andrew H. Moeller[1]

**ABSTRACT**   The gut microbiota is shaped by host metabolism. In house mice (*Mus musculus*), major urinary protein (MUP) pheromone production represents a considerable energy investment, particularly in sexually mature males. Deletion of the *Mup* gene family shifts mouse metabolism toward an anabolic state, marked by lipogenesis, lipid accumulation, and body mass increases. Given the metabolic implications of MUPs, they may also influence the gut microbiota. Here, we investigated the effect of a deletion of the *Mup* gene family on the gut microbiota of sexually mature mice. Shotgun metagenomics revealed distinct taxonomic and functional profiles between wild-type and knockout males but not females. Deletion of the *Mup* gene cluster significantly reduced diversity in microbial families and functions in male mice. Additionally, a species of *Ruminococcaceae* and several microbial functions, such as transporters involved in vitamin $B_5$ acquisition, were significantly depleted in the microbiota of *Mup* knockout males. Altogether, these results show that MUPs significantly affect the gut microbiota of house mouse in a sex-specific manner.

**IMPORTANCE**   The community of microorganisms that inhabits the gastrointestinal tract can have profound effects on host phenotypes. The gut microbiota is in turn shaped by host genes, including those involved with host metabolism. In adult male house mice, expression of the major urinary protein (*Mup*) gene cluster represents a substantial energy investment, and deletion of the *Mup* gene family leads to fat accumulation and weight gain in males. We show that deleting *Mup* genes also alters the gut microbiota of male, but not female, mice in terms of both taxonomic and functional compositions. Male mice without *Mup* genes harbored fewer gut bacterial families and reduced abundance of a species of *Ruminococcaceae*, a family that has been previously shown to reduce obesity risk. Studying the impact of the *Mup* gene family on the gut microbiota has the potential to reveal the ways in which these genes affect host phenotypes.

**KEYWORDS**   major urinary proteins, mice, knockout, gut microbiome, gut microbiota, metagenomics, taxonomy, clusters of orthologous groups (COGs), lipid metabolism, male, female, sexual dimorphism

The gut microbiota has emerged as a major modulator of host phenotypes, from metabolism (1–5) to behavior (6–8), motivating the investigation of the factors that shape the gut microbiota. Hosts can influence the taxonomic and functional compositions of the microbiota through various genetically-based physiological processes (9). Gene knockouts (KOs) allow us to test hypotheses concerning the effects of these processes on the gut microbiota. This approach has been employed to demonstrate the effects on the microbiota of innate and adaptive immune genes (10–14). However, the effects on the microbiota of non-immune genes related to host metabolism have only recently started to be investigated (15–17).

Address correspondence to Madalena V. F. Real, mv469@cornell.edu, Michael J. Sheehan, msheehan@cornell.edu, or Andrew H. Moeller, andrew.moeller@cornell.edu.

The authors declare no conflict of interest.

See the funding table on p. 11.

10.1128/spectrum.03566-23 **1**

Major urinary proteins (MUPs) are lipocalins involved in pheromonal communication (18–21), and their production represents a major metabolic investment for house mice (*Mus musculus*). In this social rodent, the *Mup* gene family has undergone an extensive parallel evolutionary expansion, accumulating 21 distinct copies in a 2.2-Mbp gene cluster on chromosome 4 (21). The genetic diversity and dynamic expression of *Mup* genes allow excreted MUPs to function as individual identifiers (22–25), conveying kinship, territory, social status, sex, reproductive state, age, health, and even diet (26–31). This communication occurs mostly through urine markings (32, 33), with MUPs constituting up to 90% of the male urinary proteome (34), a two to eight times higher protein content than females (35). *Mup* genes are also the most highly expressed genes in the liver (35), representing up to 20% of the hepatic transcriptome in mature males (36). Male MUP expression is particularly upregulated after puberty (37, 38), when social dominance is established (39, 40).

MUP production is a considerable energy investment for mice, particularly males. MUP expression is reduced under caloric restriction (41–43) and in obese and diabetic mice (41, 44). In addition to being affected by energy availability, MUPs also regulate house-mouse metabolism. Genetically obese and diabetic mice inoculated with a recombinant MUP display improved insulin sensitivity mediated by a reduction in glucose and lipid anabolism (44). Conversely, sexually mature *Mup* KO males exhibit increased anabolic phenotypes relative to wild-type (WT) individuals (45). KO males displayed higher body weight and accumulation of visceral adipose tissue than WT mice, despite lower food intake and equal energy expenditure. This metabolic shift also manifested through higher circulating levels of triglycerides, free fatty acids, and leptin and an upregulation of genes associated with lipid metabolism in KO versus WT males. These results point to the profound effect of MUPs on mouse metabolism. Given the known interactions between host metabolic function and the gut microbiota (11, 15–17, 46, 47), *Mup* expression may indirectly impact the gut microbial community. Additionally, *Mup* gene expression has been observed in the intestinal transcriptome of juvenile males (48) and in the duodenal proteome of adults (49), indicating that gut commensals could also be in direct contact with MUPs. These possible mechanisms lead us to hypothesize that deletion of the *Mup* gene family may have major effects on the house-mouse microbial community, but this has yet to be tested.

Here, we investigated how deletion of the *Mup* gene cluster impacts the gut microbiota of house mice. To answer this question, we generated a *Mup* KO line (KO; *Mup*$^{-/-}$) with CRISPR/Cas9 and crossed it with WT mice (WT; *Mup*$^{+/+}$) for multiple generations, yielding litters of mice discordant for *Mup* genotype. We then sequenced and compared the gut metagenomes of the homozygous progeny (KO versus WT). We hypothesized that sexually mature WT and KO mice would host distinct microbiotas, both taxonomically and functionally, and that the largest differences would be found in males. We found a sex-specific effect of the *Mup* KO on the microbial taxonomic and functional profiles, demonstrating that this metabolically costly gene cluster shapes the gut microbiota of house mice.

## RESULTS

### Deletion of the *Mup* gene cluster

The *Mup* gene cluster was fully deleted using CRISPR/Cas9 to cleave upstream of *Mup*4 and downstream of *Mup*21 (Fig. 1A). An individual's *Mup* status was confirmed by genotyping ear biopsies collected at weaning. Lack of MUP production was confirmed by measuring urinary MUP levels (Fig. 1B). Crosses between heterozygotes (HT) generated offspring with two copies of the *Mup* gene cluster (WTs), only one (HTs), or none (KOs) at the expected Mendelian ratio (Fig. S1).

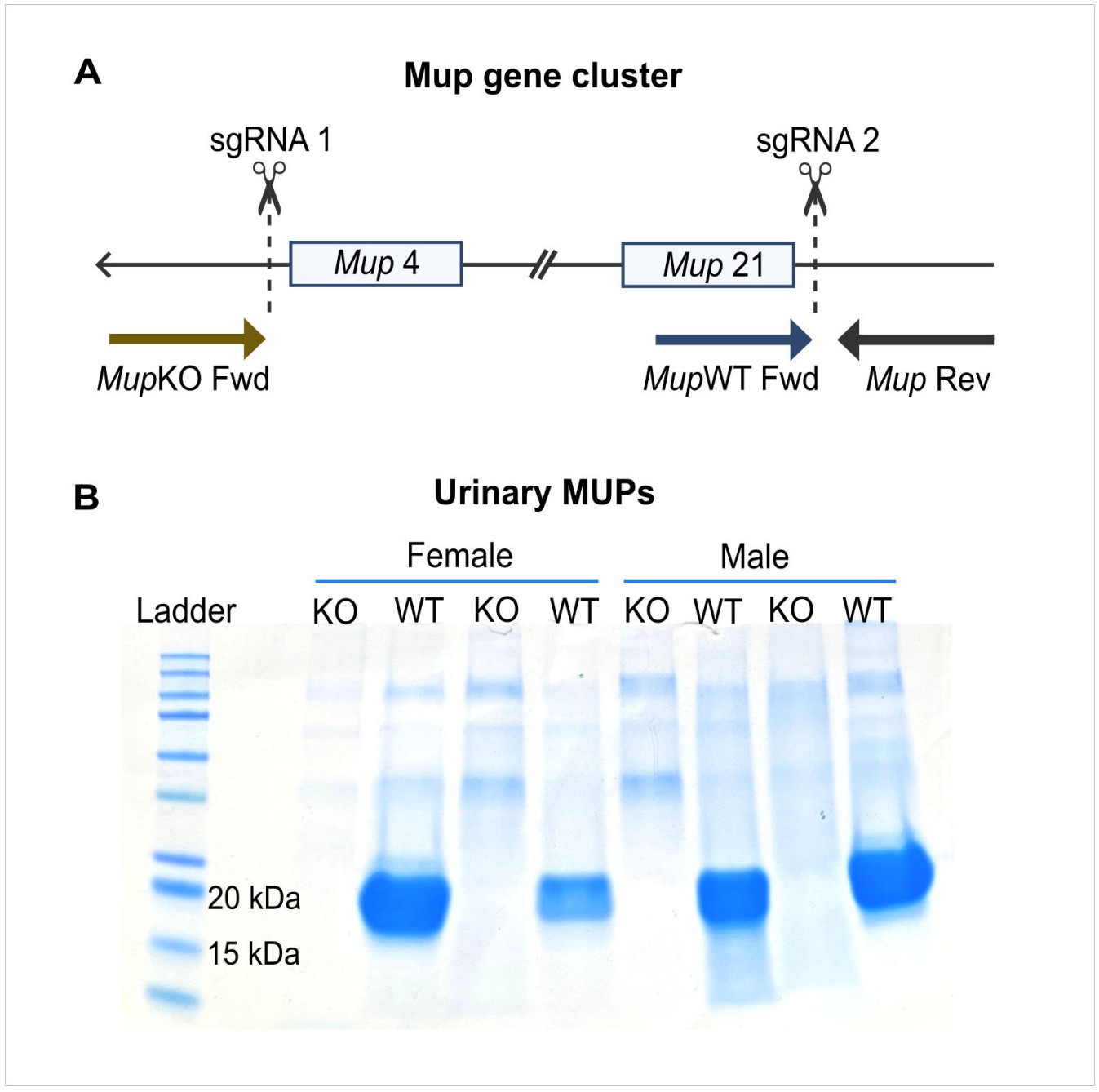

**FIG 1** *Mup* deletion and experiment timeline. (A) Representation of the 2-Mbp *Mup* gene cluster shows the CRISPR/Cas9 sgRNA target sites (dashed lines) and the primer-binding sites used for genotyping (arrows). (B) Representative SDS-PAGE stained with Coomassie brilliant blue shows urinary MUPs in WT mice (blue band at 20 kDa) and the absence of MUPs in KO individuals of both sexes ($n = 2$ mice/sex/genotype).

## Metagenomic sequencing of *Mup* WT and KO mice

MUP production is upregulated in WT males after sexual maturity (37, 38), and thus, we sampled the gut microbiota by collecting fecal pellets from 12-week-old mice (87 ± 3 days old) housed in holding cages with same-sex littermates of diverse genotypes. A total of six litters were sampled. We sequenced the fecal metagenome of WT and KO mice, yielding an average of 9.63 ± 6.05 million reads/sample post-quality control and host filtering. There were no significant differences in read count between WT and KO mice or between males and females, according to Wilcoxon tests (Holm–Bonferroni

adjusted *P* value = 0.90 and 0.19, respectively) or a linear mixed-effects model accounting for litter as a random effect (*P* value = 0.51 and 0.59, respectively). For the alpha and beta diversity analyses, all samples were rarefied to the minimum observed value of 1 million reads/sample. Functional annotation of unrarefied reads produced 478.1 ± 295.9 thousand Clusters of Orthologous Groups of proteins (COGs) per sample, which were rarefied to the minimum observed value of 84.0 thousand COGs/sample before alpha and beta diversity analyses.

## *Mup* deletion affected the taxonomic composition of the gut microbiota in males

To test the hypotheses that deleting the *Mup* gene cluster affected the composition of the gut microbiota, particularly in males, we compared the gut microbiota of sexually mature WT and KO mice. Permutational multivariate analysis of variance (PERMANOVA) based on Jaccard and Bray-Curtis dissimilarities among the samples' MetaPhlAn4 taxonomic profiles revealed that the microbiotas of male and female mice were significantly different (Table S1). Thus, we tested for significant differences between WT and KO individuals within each sex while controlling for litter effects. These analyses revealed that mature WT and KO males display significantly different microbial taxonomic compositions at the species and genus (but not family) levels based on both presence-absence (Jaccard) and abundance-weighted (Bray-Curtis) beta diversity dissimilarities, even while co-housed with same-sex littermates of diverse *Mup* genotypes (Table 1; Table S2; Fig. S2). This significant shift in the taxonomic makeup of the microbiota between WT and KO males was not driven by differences in dispersion between *Mup* genotypes, as no significant differences in the taxonomic permutational multivariate analysis of dispersion (PERMDISP) were observed (Table 1; Fig. 2B). No significant differences were observed in females (Table 1; Table S2; Fig. S2). Cumulatively, these results show that deletion of the *Mup* gene cluster caused a sex-specific directional shift in the taxonomic composition of the gut microbiota.

## *Mup* deletion affected the functional composition of the gut microbiota in males

We also tested if gene function profiles in the microbiota differed between *Mup* WT and KO mice. Using MG-RAST annotations based on COGs (Table S2), we conducted the same PERMANOVA and PERMDISP employed for the microbiota taxonomic profiles. As observed for taxonomy, the COG functional composition was significantly different between WT and KO males but not between WT and KO females (Table 1; Fig. S3). In contrast to the taxonomic results, PERMDISP indicated higher functional variation among KO males compared to WT individuals, based on Jaccard, but not Bray-Curtis, similarities (Table 1; Fig. 2B). No such differences in PERMDISP were observed in females (Table 1; Fig. 2B). These findings indicate that mature KO males exhibited both directional shifts and increased inter-individual heterogeneity in the functional composition of the gut microbiota relative to WT mice.

### Significant correspondence of taxonomic and functional profiles

On the basis of the results of PERMANOVA and PERMDISP (Table 1), we next visualized the similarities among the microbiota of WT and KO mice at both taxonomic and functional levels using Procrustes. These analyses revealed significant correspondence between taxonomic and functional profiles recovered from individual mice using Bray-Curtis but not Jaccard (Table S3). Within sexes, only in mature males did the taxonomic and functional configurations significantly match (Table S3). Plotting the superimposed taxonomic and functional compositions of mature mice revealed that samples clustered by *Mup* genotype in males but not females (Fig. 2A).

**TABLE 1** Effect of *Mup* genotype on microbial species and COG function composition[a]

| Sex | Dissimilarity | Species | | COG function | |
|---|---|---|---|---|---|
| | | PERMANOVA | PERMDISP | PERMANOVA | PERMDISP |
| Male | Jaccard | **0.037** | 0.273 | **0.021** | **0.046** |
| | Bray-Curtis | **0.019** | 0.188 | **0.012** | 0.080 |
| Female | Jaccard | 0.154 | 0.739 | 0.116 | 0.633 |
| | Bray-Curtis | 0.158 | 0.398 | 0.481 | 0.200 |

[a]Significant results are bolded (Holm–Bonferroni adjusted *P* value < 0.05).

## *Mup* deletion reduced the gut microbial diversity in males

Given the observed differences in gut taxonomical and functional profiles between WT and KO males, we investigated whether the *Mup* KO also affected taxonomic and functional alpha diversity. A linear mixed-effects model accounting for litter as a random effect indicated that KO males had lower diversity (Shannon) and evenness (Pielou) of microbial families, but not species or genera, than their WT counterparts (Table S4; Fig. S4). Females displayed no effect of *Mup* genotype on taxonomic diversity (Table S4; Fig. S4). We also observed differences in COG function diversity (Shannon but not Pielou) between WT and KO males (Table S4; Fig. S5). No such differences were observed in females (Table S4; Fig. S5). These alpha diversity results align with our findings from the beta diversity analyses, once again showing that the *Mup* gene cluster deletion affected the gut microbiota diversity of mature mice in a sex-specific manner.

## Specific microbial taxa and functions were depleted in *Mup* KO males

Given that the *Mup* KO affected the gut microbial taxonomic and functional compositions in mature males (Fig. 2), we next identified which specific taxa and/or COG

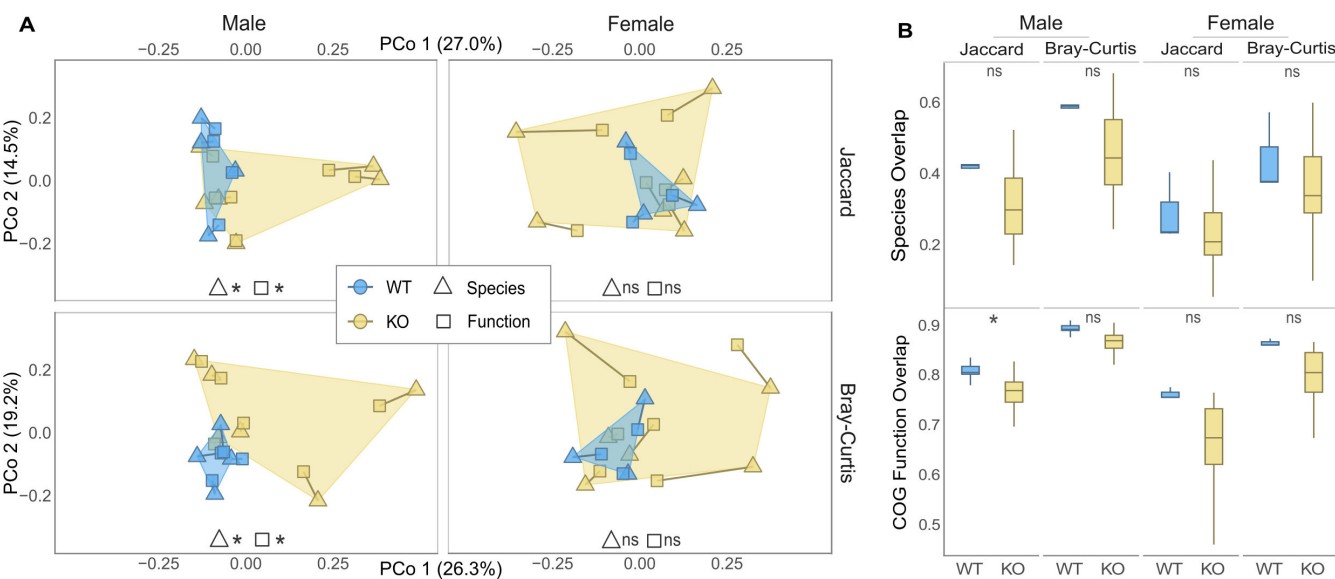

**FIG 2** *Mup* deletion significantly changes the gut microbial taxonomic and functional compositions of mature males. (A) Principal coordinate (PCo) analyses show the ordinated Jaccard (top row) and Bray-Curtis (bottom row) dissimilarities in species (triangles) and COG function (squares) composition in the fecal metagenome of sexually mature mice, superimposed with a Procrustes analysis. The ordinated points are faceted by sex and colored by genotype, showing the taxonomic and functional profiles of WT (blue) and KO (yellow) male (left column) and female (right column) mice. The percentage of variation in the taxonomic dissimilarity matrix explained by each PCo axis is enclosed in parenthesis. The PERMANOVA (center bottom) indicates whether there is a significant difference in the centroid and/or dispersion of the WT and KO groups for each dissimilarity measure and sex (* = P value < 0.05; ns = P value > 0.05). (B) Box plots show the species (top row) and COG function (bottom row) overlap in the microbiota of mature males (left column) and females (right column), using both Jaccard (left sub-column) and Bray-Curtis (right sub-column) similarities. The PERMDISP (center top) indicates whether there is a significant difference in the dispersion of the WT (blue) and KO (yellow) groups (* = P value < 0.05; ns = P value > 0.05).

functions differed between *Mup* WT and KO mice (Fig. 3; see also Fig. S6). Differential abundance analyses with ANCOM-BC2 detected a *Ruminococcaceae* species, species-level genome bin (SGB) 43260, that was present in both WT and KO mice but significantly underrepresented in KO males relative to WT males (Holm–Bonferroni adjusted *P* value < 0.001; Table S5). No taxa, including SGB43260, were differentially abundant at this significance threshold between WT and KO females (Table S5). Several COG functions were also underrepresented in KO males (Table S5). The largest fold change in abundance was observed in the sodium ($Na^+$)/pantothenate symporter, involved in the transport of pantothenate or vitamin $B_5$ (50). The most significant was component EscU of the type III secretory pathway, used by Gram-negative bacteria to inject virulence factors into host cells (51, 52). L-Asparaginase II, a periplasmic high-affinity enzyme that hydrolyzes exogenous L-asparagine into L-aspartate and ammonia (53), was also highly significant. A hypergeometric test revealed that no single COG category was particularly overrepresented among the various depleted functions in KO males (Table S6). The microbiota of KO males was instead significantly enriched in transcriptional regulators. Here, too, no functions were highly differentially abundant in the microbiota of females (Table S5). These results point to the sex-specific effects of the *Mup* gene cluster deletion on the abundance of specific taxa and functions.

## Depleted COG functions were present in the SGB underrepresented in KO males

On the basis of the results from the differential abundance analysis, which identified the *Ruminococcaceae* species SGB43260 and several COG functions that significantly depleted in KO males, we investigated whether these COG functions were present in SGB43260. We functionally annotated the 29 metagenome-assembled genomes (MAGs) from SGB43260 in the MetaPhlAn4 reference database. The genomes had an average length of 2.4 ± 0.5 Mbp of which 80.2 ± 4.8% were coding regions, with 2.2 ± 0.5 thousand protein-coding genes annotated. Some of these genes were annotated as COG functions depleted in KO males. The functions with the second and third largest fold change in abundance, asparaginase and Cu/Zn superoxide dismutase, were found in 26/29 and 16/29 genomes, respectively. $Na^+$/pantothenate symporters, the COGs with the largest fold change in abundance, were not observed in any of the genomes, although a gene identified as being part of the $Na^+$/solute symporter family was found in one of the MAGs. Although missing pantothenate-specific symporters, 24/29 MAGs had a type III pantothenate kinase, which catalyzes the first step in the pathway that converts pantothenate into coenzyme A (CoA) (54). The COG function with the most significant differential abundance result was component EscU of the type III secretory system, but no genes associated with this secretion system were found in the annotated genomes. These results indicate that the depletion in some, but not all, COG functions could be explained by the reduction in abundance of the *Ruminococcaceae* species SGB43260 in KO males.

## DISCUSSION

We found that the presence of the *Mup* gene cluster in house mice significantly altered the taxonomic and functional compositions of the gut microbiota. In accordance with our hypothesis, deletion of the *Mup* gene cluster significantly affected the gut microbiota of mature male mice, but not female mice, through a shift in composition, reduction in diversity, and depletion of microbial taxa and functions. These differences were seen even among co-housed littermates, indicating that the effects of *Mup* were robust to the homogenizing effect of microbial dispersal among animals sharing a cage (55, 56). The observation that *Mup* deletion did not change the microbial profiles of females aligns with our hypothesis that the effect of a *Mup* deletion would be stronger in male mice, given the sexually dimorphic expression pattern that this gene cluster exhibits (35, 36).

Deletion of the *Mup* gene cluster led not only to a shift in the taxonomic and functional compositions of mature males but also to an increase in inter-individual

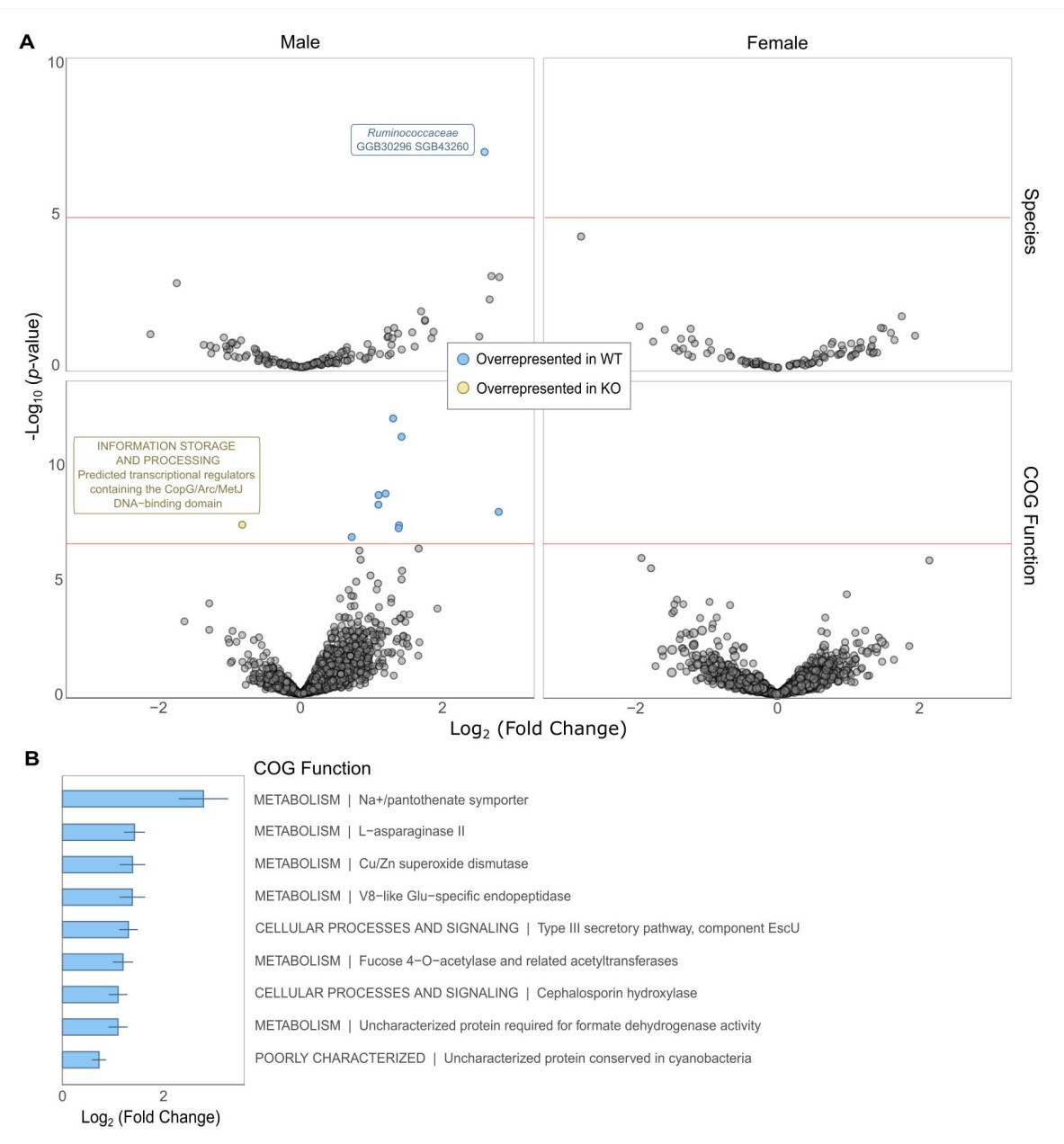

**FIG 3** *Mup* deletion significantly shifts the abundance of various microbial taxa and functions. (A) Volcano plots show the log$_2$-transformed fold change in the abundance of species (top row) and COG functions (bottom row) in the gut microbiota of mature male (left column) and female (right column) mice. ANCOM-BC2 analyses identified species and functions (points) that were significantly more abundant in WT (blue) or KO mice (yellow), labeled, respectively, with the family and species designation, and the COG category and function. Red lines mark the significance threshold (Holm–Bonferroni adjusted *P* value < 0.001). The *y*-axis indicates the −log$_{10}$ transformation of the non-adjusted *P* value. (B) Bar plots show the transformed fold change in abundance of the COG functions that were significantly overrepresented in mature WT males.

microbiota variation. The production and/or presence of MUPs at high levels may exert a parallel pressure on the microbial community of WT males, driving convergence toward a similar configuration. Mature males lacking the *Mup* gene cluster also exhibited decreased diversity and evenness in microbial families and gene functions. A reduction in microbial taxonomic and functional diversity is one of the hallmarks of obesity-related metabolic syndrome (3, 57, 58), and *Mup* KO males were previously shown to develop phenotypes associated with this syndrome, such as higher body weight, visceral adipose

tissue, and circulating levels of triglycerides, free fatty acids, and leptin (45). Cumulatively, these results are consistent with a scenario in which knocking out the *Mup* gene cluster dysregulates mouse development and/or physiology, increasing the amount of variation in functional content among individuals and reducing alpha diversity.

In addition to community-level patterns, the abundance of a *Ruminococcaceae* species, SGB43260, was negatively impacted by the deletion of the *Mup* gene cluster in males but not females. This taxon was present in several KO mice, showing that *Mup* presence is not required for colonization by this microbe. *Ruminococcaceae* have been previously associated in both mice and humans with lower body weight and reduced risk of developing metabolic syndrome (59–62). Members of this microbial family are major producers of butyrate (63), a short-chain fatty acid (SCFA) that promotes gut health (with both anti-inflammatory and antitumor properties (64)) and increases satiety (65). The depletion of this taxon in *Mup* KO males might explain the previously observed weight gain in these animals (45), although further experiments (e.g., microbiota-transplant experiments into germ-free mice) will be required to assess this hypothesis.

Several gene functions were also depleted in the gut microbiota of *Mup* KO males, particularly sodium-dependent transporters of pantothenate (vitamin $B_5$) (50). Pantothenate forms the core of CoA, an essential co-factor in cellular respiration (66) and energy metabolism, including the synthesis of SCFAs like butyrate. However, not all bacterial taxa are able to synthesize vitamin $B_5$ *de novo* (67). Several *Ruminococcaceae* species are auxotrophic for pantothenate (68, 69), relying on sodium-dependent transporters to acquire this essential vitamin and forming cross-feeding networks with pantothenate producers (67, 70). Functionally annotating the MAGs within SGB43260 did not reveal any pantothenate-specific symporters, although the MAGs did contain kinases that can utilize pantothenate in CoA biosynthesis. Other depleted COG functions, such as asparaginase and Cu/Zn superoxide dismutase, were found in the SGB43260 MAGs. These results suggest that the decrease in abundance of certain COG functions caused by *Mup* deletion could be linked to SGB43260, whereas others may not reflect changes in the relative abundance of specific taxa but instead taxonomy-independent shifts in the functional profile of the microbiota.

One potential caveat to the observed sex-dependent effects of *Mup* deletion on the house-mouse microbiota is that our study may have been underpowered to detect small effects in females. However, subsampling the WT males by randomly removing one individual and reperforming all analyses indicated that significant results were still observed in iterations based on lower sample sizes (see the Supplemental Results), suggesting that sampling effort alone cannot explain the disparity in results between males and females. Regardless, a priority for future work will be to test through expanded sampling whether WT and KO females also differ in the taxonomic and functional compositions of their microbiota. Furthermore, lineages of WT and KO mice could be housed in separate cages over longer timescales (e.g., multiple generations), as this is expected to amplify microbiota differences between *Mup* genotypes. Comparing the microbiota of immature WT and KO males and testing how the gut microbial composition changes when these animals reach puberty and start producing MUPs (37, 38) could distinguish between the effect of the *Mup* gene cluster KO and the sex-specific differences in MUP production. In the present study, the observation that adult females, which have two to eight times lower MUP content in their urine than adult males (35), did not display significant microbiota differences is consistent with the effects of sex-specific differences in MUP production.

The observed shifts in the microbiota—through a decrease in overall diversity and depletion of taxa and functions associated with host metabolic health—could in turn impact the metabolism of KO mice. It will be interesting to disentangle which aspects of the anabolic phenotype observed in *Mup* KO males are directly caused by the absence of MUP production or by the shifts in the microbial community. Reciprocally transplanting the gut microbiotas of WT and KO mice to germ-free mice with different *Mup* genotypes could elucidate which metabolic phenotypes are caused by the microbiota differences

observed here. The role of the identified *Ruminococcaceae* species could be explored by inoculating KO mice with this taxon and testing for changes in the host metabolic state. Additionally, it is worth investigating the mechanisms through which deletion of the *Mup* gene cluster affects the gut microbiota. Inoculating KO mice with recombinant MUPs (44) could help differentiate between the effects of MUP production and the role of circulating MUPs on the house-mouse gut microbial community. Our results also motivate the investigation of how the gut microbiota mediates MUP expression. Both male and female germ-free mice have reduced MUP transcription (71), which suggests that the presence of the gut microbiota or some of its members is necessary for normal MUP production. Investigating the interactions between MUPs, the microbiota, and metabolism will reveal the role of this sexually dimorphic gene on house-mouse physiology.

## MATERIALS AND METHODS

All procedures conformed to guidelines established by the U.S. National Institutes of Health and have been approved by the Cornell University Institutional Animal Care and Use Committee (protocol #2015-0060).

### Genome editing

The *Mup* gene cluster KO with CRISPR/Cas9 on FVB × B6 hybrid mice was performed by Cornell University's Stem Cell and Transgenic Core Facility. The inbred mouse strain FVB/NJ (JAX #001800) is commonly used to generate transgenics, due to its large pronucleus and litter size (72). B6(Cg)-*Tyr*$^{c-2J}$/J (JAX #000058) are C57BL/6J albino mice (73). Purified RNA (Cas9 + sgRNA; sgRNA1: GGGCCATAAGGAATGATCTTGGG; sgRNA2: GAGCTAAAGGAGACCCATATGGG) was injected into the pronucleus and cytoplasm of fertilized FVB × B6 embryos ($n = 150$). Embryos that advanced to the two-cell stage were transferred into pseudo-pregnant FVB × B6 females (20 embryos/recipient).

The resulting offspring were genotyped with Transnetyx (Cordova, TN, USA) using ear tissue samples collected at weaning. Real-time PCR was used to detect *Mup* presence, using primers that targeted the gene cluster (forward primer: ACAACCTGCCATTCTGTCT CTTAAT; reverse primer: GGCAATGAAACAAGGATTTGAGTTTTACATAT; final concentration: 900 nM). A second test confirmed *Mup* deletion by using primers flanking the gene cluster, as amplification is only possible if the 2.2-Mbp region is absent (forward primer: C AGTACTCAGGGCTTGGGATT; reverse primer: ACTGTTCTCGTGGGAATATGTATTGTGAA; final concentration: 900 nM). Successful amplification in both tests indicates HT genotype, where the *Mup* gene cluster is present in one of the chromosomes and missing from the other. WT individuals only have amplification in the detection test, while only the deletion test is successful in KO mice. The genotyping was phenotypically confirmed by measuring MUP concentration in the animals' urine with sodium dodecyl sulfate–polya-crylamide gel electrophoresis (SDS-PAGE). The gel was stained with Bio-Rad's Bio-Safe Coomassie brilliant blue, and Bio-Rad's Precision Plus Protein Kaleidoscope Prestained Protein Standards were used as a ladder.

### Animals

A breeding population was kept at a conventional mouse facility at Cornell University, Ithaca, NY, USA. Crosses between HT generated litters of WT, HT, and KO individuals. We analyzed mice from six different litters (7 ± 1 pups per litter). Mice were weaned at 3 weeks of age (24 ± 3 days) and housed with same-sex siblings of diverse genotypes (two to four animals per cage). Animals were kept in a 12-h light:12-h dark cycle, with constant room temperature and humidity (21℃, 50%). Standard chow diet and water were available *ad libitum*. Only WT and KO males and females were included in the analysis ($n = 20$; 4 WT males; 3 WT females; 6 KO males; 6 KO females).

## Microbiota analysis

Fecal samples were collected at 12 weeks of age (87 ± 3 days). Total microbial DNA was extracted using the Quick-DNA MagBead Extraction Kit (Zymo, Irvine, CA, USA) and the OT-2 liquid handling robot (Opentrons, New York, NY, USA). Library preparation followed the Hackflex protocol (74) using the same robot. Libraries were sequenced on an Illumina NextSeq 2000 (Biotechnology Resource Center, Cornell University, Ithaca, NY, USA).

The metagenomic data were quality controlled with FastQC (v0.12.1) (75), followed by trimming of the Illumina adapters (GATCGGAAGAGC) with Cutadapt (v4.1; setting: "--minimum-length 1 --nextseq-trim 20") (76), and removal of host reads with Bowtie2 (v2.5.1; setting "--very-fast"; host reference genome: GRCm39 GCF_000001635.27) (77). The remaining reads were taxonomically profiled with MetaPhlAn4 (v4.0.6) (78) and functionally annotated with COGs using MG-RAST (v4.0.3) (79). The genomes included in SGB43260 from the MetaPhlAn4 reference database were functionally annotated with Bakta (v1.7.0) (80).

All data analyses were conducted in R (v4.2.2). Before measuring alpha and beta diversity, the library size was normalized by randomly subsampling sequences to 1 million reads/sample for the taxonomic data and 84 thousand COG counts/sample for the functional data. The Shannon diversity index, observed richness, and Pielou's evenness metrics were measured using the microbiome R package (v1.20.0) (81). The Jaccard and Bray-Curtis dissimilarities between samples were calculated with phyloseq (v1.42.0) (82). All plots were created with ggplot2 (v3.4.1) (83). All figure panels were assembled using Inkscape 1.2.

## Statistical analysis

Data are presented as mean ± standard deviation. Group means were compared with the Wilcoxon signed-rank test. A linear mixed-effects model assessed the effect of genotype on microbiota diversity, while controlling for litter as a random effect (via lmerTest v3.1 (84)). The factors (genotype, sex, and/or litter) that explained the microbiota dissimilarity matrix were tested by running a PERMANOVA (85) via adonis2 and a PERMDISP (86) via betadisper (vegan v2.6 (87)). Differentially abundant taxa and functions were identified with ANCOM-BC 2 (88). Enrichment/depletion in COG categories within differentially abundant functions was tested with a hypergeometric test via phyper (stats v4.2.2). Throughout the analyses, a Holm–Bonferroni adjusted $P$ value of <0.05 was considered statistically significant, except for the ANCOM-BC 2 analyses, where a Holm–Bonferroni adjusted $P$ value of <0.001 was used.

## ACKNOWLEDGMENTS

We acknowledge Cornell University's Stem Cell and Transgenic Core Facility for generating the *Mup* knockout mice and Cornell University's Biotechnology Resource Center for the Illumina NextSeq 2000 sequencing. We thank the Segata Lab for sharing the MAGs from SGB43260. We also thank Tess Reichard for her help with the urine collections and the SDS-PAGE and Dr. Weiwei Yang and Dr. Jon Sanders for their assistance with the mouse fecal DNA extractions, DNA library preparation, and submission of the DNA libraries for sequencing.

Funding was provided by the National Institutes of Health (NIH) grant R35 GM138284-01 (A.H.M.) and a pilot grant award under NIH Animal Models for the Social Dimensions of Health and Aging Research Network R24 AG065172 (M.J.S.). NIH had no role in study design, data collection and interpretation, or the decision to submit the work for publication. The authors declare no competing interests.

M.V.F.R.: conceptualization, methodology, investigation, formal analysis, visualization, writing – original draft, and writing – review & editing; M.S.C.: methodology and investigation; M.J.S.: conceptualization, methodology, investigation, writing – review & editing, and funding acquisition; and A.H.M.: conceptualization, methodology, investigation, formal analysis, visualization, writing – original draft, writing – review & editing, and funding acquisition.

## AUTHOR AFFILIATIONS

[1]Department of Ecology and Evolutionary Biology, Cornell University, Ithaca, New York, USA

[2]Department of Neurobiology and Behavior, Cornell University, Ithaca, New York, USA

## AUTHOR ORCIDs

Madalena V. F. Real ⓘ http://orcid.org/0000-0003-0550-8252
Michael J. Sheehan ⓘ http://orcid.org/0000-0002-3949-7873
Andrew H. Moeller ⓘ http://orcid.org/0000-0002-8377-4647

## FUNDING

| Funder | Grant(s) | Author(s) |
| --- | --- | --- |
| HHS | National Institutes of Health (NIH) | R35 GM138284-01 | Andrew H. Moeller |
| HHS | National Institutes of Health (NIH) | R24 AG065172 | Michael J. Sheehan |

## AUTHOR CONTRIBUTIONS

Madalena V. F. Real, Conceptualization, Formal analysis, Investigation, Methodology, Visualization, Writing – original draft, Writing – review and editing | Melanie S. Colvin, Investigation, Methodology | Michael J. Sheehan, Conceptualization, Funding acquisition, Investigation, Methodology, Writing – review and editing | Andrew H. Moeller, Conceptualization, Formal analysis, Funding acquisition, Investigation, Methodology, Visualization, Writing – original draft, Writing – review and editing

## DATA AVAILABILITY

All FASTQ sequence data and associated metadata have been deposited in NCBI's Sequence Read Archive under BioProject accession no. PRJNA995784. All the analyses conducted in this manuscript and supplemental data tables are available online (https://github.com/CUMoellerLab/Real_etal_2023_MUPKO).

## ADDITIONAL FILES

The following material is available online.

### Supplemental Material

**Supplemental material (Spectrum0356-23-s0001.pdf).** Supplemental results; Figures S1 to S6; Tables S1 to S8.

### Open Peer Review

**PEER REVIEW HISTORY (review-history.pdf).** An accounting of the reviewer comments and feedback.

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
