## [Reviewer comments · Microbiology Spectrum]

Microbiology Spectrum

Major urinary protein (*Mup*) gene family deletion drives sex-specific alterations in the house mouse gut microbiota

Madalena Real, Melanie Colvin, Michael Sheehan, and Andrew H. Moeller

Corresponding Author(s): Andrew H. Moeller, Cornell University

Review Timeline:

Submission Date:

October 4, 2023

Accepted:

November 23, 2023

Editor: Jan Claesen

Reviewer(s): The reviewers have opted to remain anonymous.

Transaction Report:

DOI: <https://doi.org/10.1128/spectrum.03566-23>

Re: Spectrum03566-23 (Major urinary protein (*Mup*) gene family deletion drives sex-specific alterations on the house mouse gut microbiota)

Dear Dr. Andrew H. Moeller:

Thank you for carefully addressing the Reviewer comments and suggestions. I think the revised version of your manuscript looks great and would like to congratulate you on its acceptance for publication in Spectrum.

Your manuscript has been accepted, and I am forwarding it to the ASM production staff for publication. Your paper will first be checked to make sure all elements meet the technical requirements. ASM staff will contact you if anything needs to be revised before copyediting and production can begin. Otherwise, you will be notified when your proofs are ready to be viewed.

Sincerely,
Jan Claesen
Editor
Microbiology Spectrum